# Supporting Home-Based Self-Regulated Learning for Secondary School Students: An Educational Design Study

**Mingzhang Zuo [1], Qifang Zhong [1,2,*], Qiyun Wang [3], Yujie Yan [1], Lingling Liang [1], Wenjing Gao [1,4] and Heng Luo [1,*]**

[1] Faculty of Artificial Intelligence in Education, Central China Normal University, Wuhan 430079, China; mzzuo@mail.ccnu.edu.cn (M.Z.); yanyujie@mails.ccnu.edu.cn (Y.Y.); rebecca1004@mails.ccnu.edu.cn (L.L.); wenjinggao@mails.ccnu.edu.cn (W.G.)

[2] Physics Teaching and Research Group, Wuhan NO. 3 Boarding School, Wuhan 430050, China

[3] Learning Sciences and Assessment Academic Group, National Institute of Education, Nanyang Technological University, Singapore 637616, Singapore; qiyun.wang@nie.edu.sg

[4] ICT Teaching and Research Group, Tsinghua University High School-Changping, Beijing 102200, China

* Correspondence: jennyzqf@mails.ccnu.edu.cn (Q.Z.); luoheng@mail.ccnu.edu.cn (H.L.)

**Abstract:** The implementation of home-based learning for secondary school students faces challenges such as weakened supervision, a lack of prior online learning experience, and low self-regulated learning (SRL) skills. To address this, we propose an implementation mechanism to help teachers develop students' SRL skills in home-based learning environments. After three iterations of design, implementation, and evaluation, following the educational-design research approach, the proposed implementation mechanism was empirically validated and refined. The results confirmed the feasibility and effectiveness of the proposed framework, one which integrates strategies of goal setting and planning, self-monitoring, and self-evaluation. We also demonstrated that the designed implementation mechanism, which comprises the four components of sequence, resource, activity, and incentive, helped students master SRL skills and improve nonacademic performance. Lastly, we identified seven design principles that can guide educators in the adoption of similar practices to develop students' SRL skills, particularly for future flexible and smart learning scenarios. These principles emphasize the motivational, sequential, social, and instrumental aspects of instructional design, and call for parental involvement and a flexible mindset during implementation. The paper ends with a discussion of several limitations regarding sample representativeness and data diversity that should be noted when interpreting the study results.

**Keywords:** self-regulated learning; home-based learning; implementation mechanism; educational-design research; secondary education

## 1. Introduction

During the COVID-19 pandemic, many schools were closed, and students had to attend classes online. At its worst, over 194 worldwide schools were fully closed, and almost 1.6 billion primary and secondary school students studied at home [1]. As this was an emergent event, most teachers, students, and parents were not fully prepared. Different from college students or adult learners, K–12 students often lack online learning experience [2,3]. In addition, unlike face-to-face learners, home-based students are not only physically separated from teachers and classmates [4], but also psychologically separated in terms of understanding and meaning-making [5]. At the same time, teacher supervision tends to be weakened in online learning, and students have less contact with teachers and classmates [6]. Such factors hinder students' adaptation to online learning, and they face challenges in home-based online learning as a result.

However, learning online also gives students opportunities to develop self-regulated learning (SRL) skills, which are related to the development of lifelong learning skills [7,8]. Many studies have indicated that high-achieving students exhibit significantly greater use of SRL skills [9,10] and that SRL is an important component of successful online education [11,12]. However, studies have also found that young or less skilled students have difficulty regulating their learning processes in online learning environments [13,14]. This could be particularly true for primary and secondary school students who attended full-time online courses from remote sites during COVID-19, possibly with slow Internet connections [15]. Worse still, students in China had not received sufficient training in SRL to help them quickly adapt to online learning. Data indicated that over 45% of students experienced difficulties when taking online courses remotely [16]. Additionally, students with low self-regulatory skills were found to frequently suffer from issues such as attention deficiency, distractions, and a lack of motivation and self-discipline while learning online [17,18]. Thus, developing students' SRL skills can help them adapt to home-based learning in times of COVID-19 [19], and to flexible or hybrid learning after the pandemic. Additionally, improved SRL is known to increase students' self-efficacy and learning efficiency [15,20], which will benefit their development in the long term.

Although there is extensive research on improving students' SRL skills in online learning, most studies have focused on either higher education [7,21] or K–12 students in blended-learning settings [22,23]. Furthermore, previous studies have quantitatively examined the relationships between SRL and learning satisfaction, interaction, effectiveness, and other variables [20,24,25], as well as the effects of a specific strategy, discipline, or software [26–29]. However, few studies have investigated the process of integrating various strategies into teaching practice to support secondary school students' SRL in online learning environments [30,31]. Therefore, this study explored how to integrate SRL strategies—such as goal setting and planning, self-monitoring, and self-evaluation—into the online learning practices of secondary school students. Accordingly, adopting the educational-design research approach, we documented the formation and refinement of an SRL implementation mechanism used in an experimental class in Wuhan, China, where COVID-19 was first reported. We also investigated students' experiences and perceptions of the SRL process. Specifically, we aimed to answer the following research question:

What are the characteristics of a feasible and effective implementation mechanism for developing secondary school students' SRL skills in a home-based learning environment?

Here, feasible means that the SRL support is acceptable to both teachers and students, is practical to implement, and can be integrated into existing classrooms to meet the needs of both teachers and students [32,33]. Effective pertains to the perceived usefulness among students [34] of SRL strategies as well as the improvement in SRL skills and performance, determined using multivariate data analysis, including both qualitative and quantitative analysis.

## 2. Conceptual Framework

Our conceptual framework for designing a practice to develop secondary school students' SRL skills was informed by a cyclical-phase model of SRL (the core theory layer), the key objectives of SRL (the second theory layer), and four initial design principles of SRL (motivation, tangible records, family–school co-education, and social interaction). In this way, SRL theory could be actualized in practice (Figure 1). To maximize the effect of practice based on SRL theory, we carried out our practice as an implementation mechanism with four components: sequence, SRL activity, resource, and incentive. In turn, the practice results verified and refined the theory.

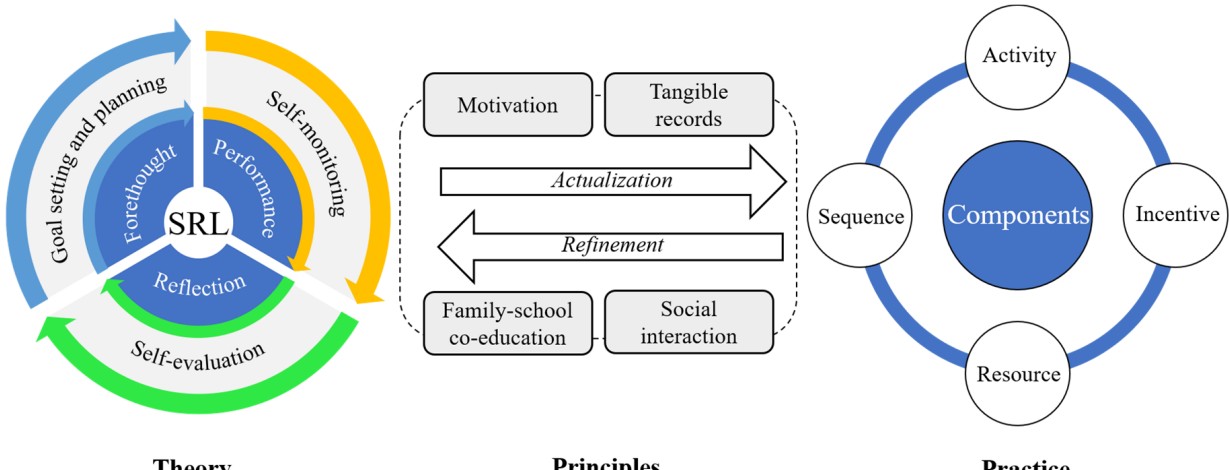

**Figure 1.** Design framework for developing school students' self-regulated learning (SRL) skills.

## 2.1. Three-Cycle Phased Model of Self-Regulated Learning

SRL is conceptualized as learning or capability that students engage in via their own learning processes—metacognitively, motivationally, and behaviorally—aiming toward the attainment of learning goals [8,31]. Studies have identified several models of SRL that are either cyclical or dynamic and are composed of various phases and subprocesses [31,35].

The most influential SRL model is Zimmerman's three-cycle, interdependent phase model, composed of forethought, performance, and reflection [35,36]. It is commonly used in computer-based learning environments [37]. In the forethought phase, students make preparations to learn; this involves motivational beliefs such as self-efficacy, which stimulate their learning [38]. Then, in the performance phase, they use self-monitoring and control their performance to optimize learning. Finally, in the reflection phase, students give personal explanations of their learning outcomes [39]. In this approach, however, there is no strong assumption in favor of structuring the phases hierarchically or linearly; for example, that previous phases must always occur before the next phases [40]. Studies have revealed that skilled, self-regulated students often spend much time thinking about and planning their learning during the forethought phase [31] and reflecting on themselves in the reflection phase [41]. However, students with low SRL capability often lack such awareness and skills [3].

## 2.2. Key Objectives of Self-Regulated Learning

There is evidence that self-regulation skills can be acquired by training via specific SRL strategies, such as self-efficacy, metacognitive, effort-regulation, and time-management strategies, which have been identified as significant factors in the effectiveness of online learning [7,21,42,43]. Among them, metacognitive skills, which are students' internal guides of their awareness of cognitive processes, play important roles in students' planning and organizing in the forethought phase, observing and monitoring in the performance phase, and self-evaluating in the reflection phase [44]. Therefore, a key objective of SRL in this study is the development of metacognitive skills, which generally include three processes: goal setting and planning, self-monitoring, and self-evaluation [7,45].

### 2.2.1. Goal Setting and Planning

Goal setting involves students setting specific outcomes or subgoals that can guide their learning performance [36,45]. It is generally combined with a planning process that plans the sequence, time, and manner of activities to achieve the set goals [46]. Goals can motivate and direct students' attention, effort, behavior, and activity selection [30,47]. Studies have revealed that students with specific proximal goals and appropriate planning usually achieve greater success in the target area and display higher SRL skills [8,48], even

in online learning environments [49,50]. However, secondary school students usually lack the skills necessary for setting specific, measurable, achievable, realistic, and time-based goals (SMART principles), which are criteria developed based on the theory of goal setting [51].

### 2.2.2. Self-Monitoring

In self-monitoring, students informally track specific aspects of their learning performance process, environment, and outcomes [38]. Research shows that self-recording is an effective self-monitoring technique that can capture important information at the point where it occurs and increase the timeliness, reliability, and accuracy of feedback [36] to enhance students' self-control and facilitate self-reflection [38]. Additionally, previous studies have revealed that keeping SRL diaries and using event questions or prompts regarding students' tasks are effective ways to help students monitor and improve their SRL skills [25,52].

### 2.2.3. Self-Evaluation

Self-evaluation involves comparing one's own performance with some quality criteria or goals for progress [8]. Opportunities for students to self-evaluate have beneficial effects on their learning progress [53,54]. Activities that monitor the learning process can help students self-evaluate and perform better in the next stage [30]. However, students lacking specific forethought goals usually fail or just use social comparison with classmates to self-evaluate [38]. In such cases, guiding students to self-evaluate using their SRL diaries and specific forethought goals is especially important in practice.

However, approaches to developing SRL skills should be integrated and structured when applied in practice, in order to maximize skill enhancement [30,44]. Thus, this study explored how to integrate SRL strategies into practice to develop secondary school students' SRL skills, with a focus on metacognitive skills in online settings.

### 2.3. Initial Design Principles of Self-Regulated Learning Development

To maximize students' acquisition of SRL skills, we established four initial design principles for SRL development that actualize theory into practice and pedagogical instruction (Table 1). Since acquiring self-regulatory skills involves time and effort, motivation is commonly regarded as a prerequisite for effectively utilizing skills such as the monitoring and regulation of studying [36,50]. Students with a positive motivational profile usually achieved higher scores in monitoring accuracy and learning outcomes compared to those with a negative motivational profile [55]. Thus, stimulating positive motivation is the basis for encouraging students to engage in and sustain the learning of SRL skills [50,56]. Self-recording serves as an effective technique for self-monitoring, promoting students' self-control and reflection [36]. Positive effects were observed when employing a discipline-independent approach, combined with SRL training and the utilization of daily recordings [25]. Hence, we used tangible records as a tool for students to practice skills by adding planning, self-monitoring, and self-reflection prompts [57,58]. Social interaction provides a way to obtain feedback and help improve skills in online learning environments [37,59,60]. A previous study showed that SRL training, particularly when supplemented with peer feedback, yielded positive outcomes [61]. Family–school co-education is a booster for skills learning because students spend most of their time at home and parents play an important role in the growth of SRL skills [58]. Studies indicate that parental involvement in education and building a family–school co-educative environment characterized by mutual trust, collaboration, respect, complementary expertise, and shared goals can help improve students' self-regulation skills [62,63].

**Table 1.** Four initial design principles for developing secondary school students' SRL skills.

| Design Principle | Description | Main Reason | Supporting Literature |
|---|---|---|---|
| P1. Stimulate motivation. | Stimulate external and internal motivations to promote the more active and sustainable engagement of students' SRL. | Extra time and effort need to be invested. | Pintrich [56]; Wang et al. [50] |
| P2. Implement individual tangible records. | Use individual tangible records to plan and monitor students' own learning processes. Embedded guides and prompts are designed to facilitate self-reflection. | Skill learning requires practice and scaffolding. | Broadbent et al. [25]; Zheng [57]; Zimmerman et al. [58] |
| P3. Create opportunities for social interaction. | Create online communication opportunities to obtain peer and instructor feedback on SRL processes to facilitate further engagement with and understanding of SRL skills. | Learning requires feedback and interaction. | Hattie and Timperley [59]; Zhu [60]; Zimmerman and Tsikalas [37] |
| P4. Build a family–school co-education environment. | Build a good family–school co-educative environment to promote timely communication between families and schools and create opportunities for parents to participate in children's SRL and transition them from coregulation to self-regulation. | Parental involvement facilitates skill improvement. | Minke and Anderson [62]; Ormrod [63]; Zimmerman et al. [58] |

## 3. Methods Employed

### 3.1. Research Design

This study adopted the educational-design research approach (also called design-based research), which aims to produce a high-quality artifact through multiple iterations and generate contextually sensitive design principles [64]. This approach emphasizes close collaboration between researchers and practitioners in real-world settings [65]. Hence, the principal investigator (PI) of this study monitored what happened when students used SRL strategies to learn online at home and discussed how to better implement SRL activities in class with the instructor. In addition, three research assistants took observational notes when the students were learning online and participated in class SRL activities to guide students in using SRL strategies.

Figure 2 depicts the research process. First, we analyzed problems related to needs and context and reviewed the literature to develop a conceptual framework and initial design principles. Then, the initial instructional design was developed, implemented, and evaluated. After multiple iterations of these processes, the problem was eventually solved to a certain extent. This study underwent three iterations of prototyping. Validated SRL strategies and refined instructional design principles were generated, and the implementation mechanism for practice was refined at the end of the research.

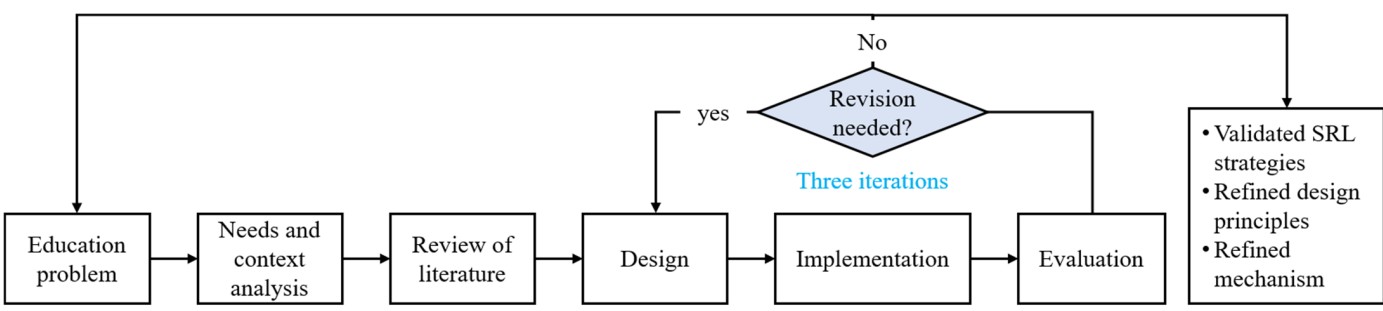

**Figure 2.** Educational-design research process.

### 3.2. Research Context

3.2.1. The Research Site

This study was conducted in a secondary school in Wuhan, China, where students had begun home-based learning in mid-February 2020. Since no suitable teaching platform was in place, the school temporarily used CCtalk (https://school.cctalk.com/, accessed on 28 January 2024) for real-time interactive teaching. Using this platform, the students could schedule classes in a virtual classroom. The real-time communication software QQ v9.2.2 (https://im.qq.com/, accessed on 28 January 2024) was used as their home–school communication platform, where the students or parents could contact teachers individually to ask questions or submit homework.

3.2.2. The Education Problem

The school was a typical traditional three-year middle school in China. Such schools have a higher rate of high school enrollment compared to other schools in the local area of the same educational stage, and 13 subject courses (e.g., moral education, music, and art) must be offered in eighth grade [64]. At the end of the third year, most students take the High School Entrance Examination, which covers six subject courses. Consequently, students' SRL skills are not usually emphasized, and learning mostly depends on face-to-face teacher supervision [28,42]. During the pandemic, the school only offered five main subject courses, with no SRL instruction, for seven periods per day (45 min each). Usually, the morning session was for new-lesson teaching, and the afternoon session was for subject tutoring, in which students completed exercises and could ask the teacher questions. However, the main online instructional format was still lecture-based, and less than 9% of students engaged in online interaction [16]. This lack of emphasis on SRL and long-term lecture-based learning resulted in students having weak SRL competency, as evidenced by frequent distraction and procrastination and increased computer-game use [19,66]. Some teachers wanted to enhance students' SRL skills but did not know how [17].

3.2.3. Participants

The participants included 58 eighth-grade students (35 boys and 23 girls). Most lived in urban areas when the epidemic began, but several lived in rural areas where Internet connections were weak, and they had to rely on mobile phones for online courses. Moreover, almost all of the students did not have physical textbooks or learning materials, and only about one-third had printers at home. Importantly, it was noted that the students generally lacked SRL skills, as evidenced in their inability to submit assignments on time and their lack of goal planning and self-discipline. Therefore, insufficient resources, long periods of screen-time, weak supervision, and poor SRL ability were practical problems facing online learning that caused concern in parents.

The instructor—who was also the head teacher, responsible for class management and family–school communication [67]—had more than 25 years of teaching experience in physics. He was troubled by the prospect of having the students learn more autonomously at that time. Thus, we embedded the training and guidance for SRL activities into some physics courses and the class meeting period, which was organized by the instructor, combined with personalized online guidance after online classes.

### 3.3. Data Collection

As shown in Table 2, we collected four main types of qualitative and quantitative data to gather evidence to evaluate the instructional design. The first source of data comprised observations from both field notes and video recordings used to capture the online teaching and learning processes during class meeting time. The second source involved artifacts submitted by students, which included monitoring journals, goal-setting documents, and planning forms. This allowed us to analyze how students used monitoring journals and engaged in SRL skill-learning processes. The third source comprised interview transcripts from students, teachers, and parents. We purposefully chose nine students (5 boys and

4 girls) from different groups and levels for a semi-structured interview to gather in-depth information about their experiences and attitudes toward the mechanism and strategies. Finally, the fourth source involved a survey, with both parents and students being surveyed to measure their participation in family–school co-education.

**Table 2.** Data collection.

| Data Collection Methods | Data Source/Measurement | Main Purpose | Data Collected |
|---|---|---|---|
| Observation | Online teaching and learning process during class meeting time, class home–school group, and each group's online discussion. | Capture class arrangement, teacher's instruction, and students' behavior and experiences. | Field notes, discourse data on the social media platform, and 13 video recordings. |
| Artifact collection | Materials used in the class or submitted by students. | Analyze how students used monitoring journals and their SRL skill-learning processes. | 1701 pictures of monitoring journals, goal-setting and planning forms, class rules, etc. |
| Semi-structured interviews | Nine purposefully chosen students (five boys, four girls) from different groups and levels, the head teacher, and three parents, all interviewed online. | Gather in-depth feedback and comments on the main SRL strategies, including students' learning experience, benefits, and challenges. | Students: 33,495-word interview transcript (three rounds of interviews); teachers: 4610-word interview transcript; parents: 15,116-word interview transcript. |
| Feedback questionnaire | All parents and students; five five-point Likert scale items (1: strongly disagree; 5: strongly agree) and open-ended questions. | Gather feedback on the design for family–school co-education aspects and perceptions of monitoring recordings and group communication. | 56 valid parent questionnaires and 51 valid student questionnaires; 31 comments and 14 relevant suggestions. |

*3.4. Data Analysis*

We carried out data analysis concurrently with data collection. Researchers' field notes, students' journals, and interview transcripts were analyzed to evaluate how the instructional design progressed, according to the rubric for iteration evaluation, which was based on the criteria of feasibility and effectiveness. As shown in Table 3, the feasibility criterion consists of the two elements of perceived ease of use—namely, the extent to which an individual believes that using a system will be effortless [34]—and appeal. The effectiveness criterion includes the two elements of perceived usefulness—namely, the extent to which an individual believes that using a specific system will enhance job performance [34]—and the improvement of SRL skills and performance. For example, in the evaluation of the first iteration, students said the monitoring form "was extremely useful", "made things clear", "increased efficiency", and "helped me recognize my procrastination", thus fulfilling the intended purpose with excellent perceived usefulness. However, several students used language such as "unaccustomed", "dislike", "scattered", and "easy to lose", indicating that perceived ease of use and appeal needed improvement. Then, the research team would determine which instructional design to iterate and adjust based on the scoring results of the rubric. Multiple analytical methods were employed based on the types of data being examined. These included thematic analysis for qualitative data, the document analysis of journals, and statistical analysis for quantitative data, which allowed for data triangulation and ensured a more comprehensive analysis.

**Table 3.** Rubric for the iteration evaluation of design principles.

| Criteria | Element | Excellent (3) | Satisfactory (2) | Needs Improvement (1) |
|---|---|---|---|---|
| Feasibility | Perceived ease of use<br><br>Appeal | It is convenient to use and little training is needed. Parents, students, and teachers all like this form. | Some simple training is required, but it is feasible. Parents, students, and teachers can find this form acceptable. | It is inconvenient to use or requires long-term guidance. Parents, students, and teachers are resistant to this form. |
| Effectiveness | Perceived usefulness | Small time investment; all intended purposes are fulfilled. | Some time investment and some intended purposes are fulfilled, but investment is not proportional to the intended purpose. | A lot of time investment, but few intended purposes are fulfilled. |
| | SRL skills and performance improvement | Students often use strategies; performance shows significant progress. | Students occasionally use strategies; performance shows some progress. | Students rarely use strategies; no obvious progress in performance. |

### 3.4.1. Thematic Analysis

Thematic analysis is a method used to develop, analyze, and interpret patterns across a qualitative dataset [68]. We used a data-driven inductive approach to identify themes from the dataset [69], which included interview transcripts for students (coded S1–S9), the teacher (coded T), and parents (coded Pi1–Pi3), as well as parents' comments in the open-ended questions (coded P1–P31). We employed a six-step thematic analysis procedure: become familiar with the data, generate initial codes, generate initial themes, develop and review themes, define and name themes, and produce the report [70].

During the data collection and review process, we familiarized ourselves with the dataset content by observing SRL activities and taking field notes. For interview data collection, at least two researchers participated, with one asking questions based on the interview outline while the other recorded and supplemented questions as necessary. Transcriptions of the interviews were promptly created in a Microsoft Word document. Then, initial codes were generated to condense the data, with a focus on evaluating instructional design principles in students' SRL processes. The coding process involved two cycles of coding, including methods such as structural coding, in vivo coding, magnitude coding, and evaluation coding (Appendix A Table A1) [71]. A total of 897 nodes, 29 first-level codes, and 77 s-level codes emerged using NVivo 12. Selected quantified coding results can be found in Appendix A Table A2.

In the second cycle of coding, pattern coding and elaborative coding techniques were used [71]. Pattern coding involved combining similar codes to create meaningful units of analysis, such as merging codes like FOUND TIME LIMITED, "OFTEN IN A DAZE", and ESTIMATED TIME VS ACTUAL TIME into the pattern code BETTER REFLECTION. Elaborative coding utilized theoretical constructs to refine and elaborate on thematic findings. For example, during the second iteration, students mentioned "SUPERVISE", "SHARE", "HELP EACH OTHER", and "INCREASE EFFICIENCY" when talking about the effect of weekly group discussions. One student mentioned NOT ALONE during the third iteration, which expanded the thematic finding that the design of the group online discussion was useful. The next process included generating, developing, and defining themes based on initial codes. Collaboration between two researchers and visual analysis using thematic maps aided in refining and capturing the most significant elements of the data, particularly those related to the four instructional design principles. Finally, the thematic analysis concluded with the reporting of the results.

### 3.4.2. Document Analysis

Document analysis is a structured procedure for examining or evaluating documents to extract meaning, enhance comprehension, and generate empirical knowledge in qualitative research. This usually involves three iterative activities—skimming, reading, and interpretation—fusing elements of content analysis and thematic analysis [72]. Students'

monitoring journals (coded J1–J58) were the main documents that provided supplementary data to track students' changes and development in SRL. Two researchers separately coded the content and recorded the corresponding frequencies to better understand students' strategies. For example, based on students' records of distractions, we coded EATING, OTHER PERSON, MOBILE PHONE, TV, and MUSIC, among others, and recorded their frequencies to analyze the main distractions and changes over time.

### 3.4.3. Statistical Analysis

SPSS v. 23 was used to analyze the data, including students' and parents' feedback questionnaires. We performed basic descriptive statistics on the mean and variance values of the questionnaires.

### 4. Initial Design of the Instructional Implementation Mechanism

Given that the theories of SRL strategies are highly varied, there is a lack of systematic teaching interventions that can be implemented in practice. Hence, integrating the four initial design principles synthesized from the literature and the research context, we developed an initial design for an instructional implementation mechanism for developing secondary school students' SRL skills in online settings (Figure 3).

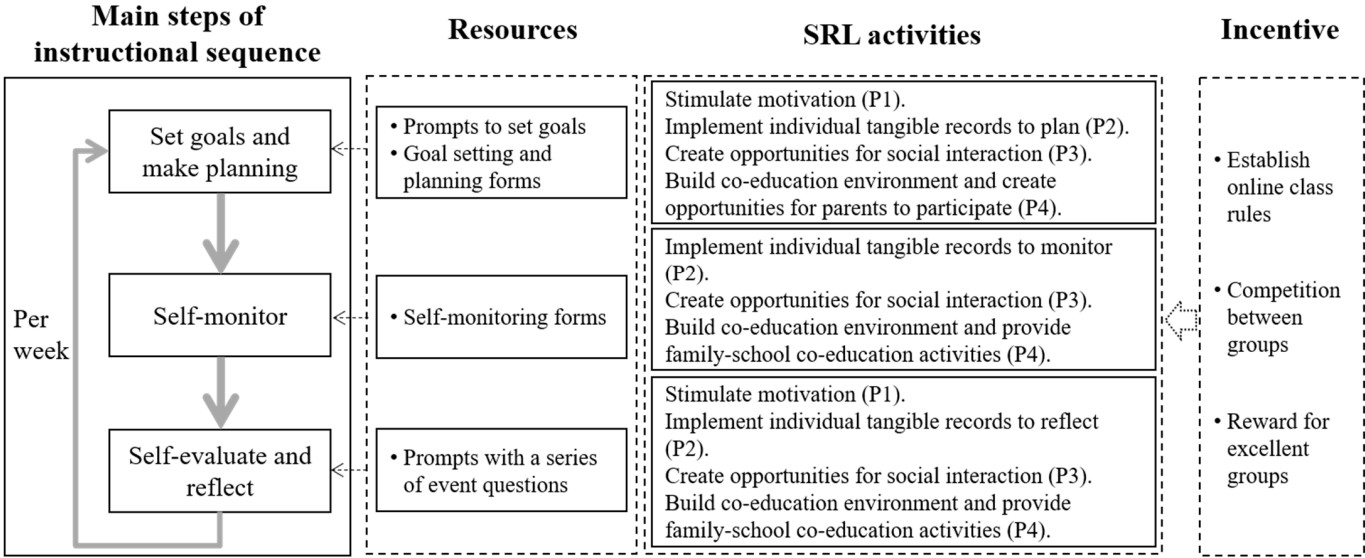

**Figure 3.** Initial design of the instructional implementation mechanism for the first iteration.

The instructional design of the implementation mechanism consists of four components, as shown in Figure 1: incentive, sequence, resource, and activity. Students and teachers negotiated the incentives to stimulate the students to participate in SRL activities. The main steps of the instructional sequence included the main SRL strategies: goal setting and planning, self-monitoring, and self-evaluation. The resource column elaborates on each step, along with the specific materials used to support students in mastering SRL skills. The SRL activities column informs students about the instructional activities (as listed in Table 1) within each sequential step. The instructor implemented the first round based on this plan.

### 5. Results

We report the results in three iterations, following the process for developmentally evaluating educational-design research.

*5.1. First Iteration*

5.1.1. Implementation

The first iteration lasted approximately 4 weeks, as shown in Figure 4. Through negotiations among students, instructors, and parents, the online class rules were established in the first week. Students were divided into 14 groups, and each group, comprising four students at different levels and a leader, could use the QQ group to collaborate and communicate online. Their parents and a research assistant also joined the QQ group to learn about the group's dynamics. Each student could earn extra points for the group by actively participating in learning activities using a self-monitoring form. By contrast, nonparticipation or a lack of seriousness would result in points being deducted from the group. The group leader was responsible for supervising and urging the group members to participate in activities. Group scores were tallied once every 2 weeks, and then the scores were reset. The three groups with the highest scores received generous rewards, which the students were excited about.

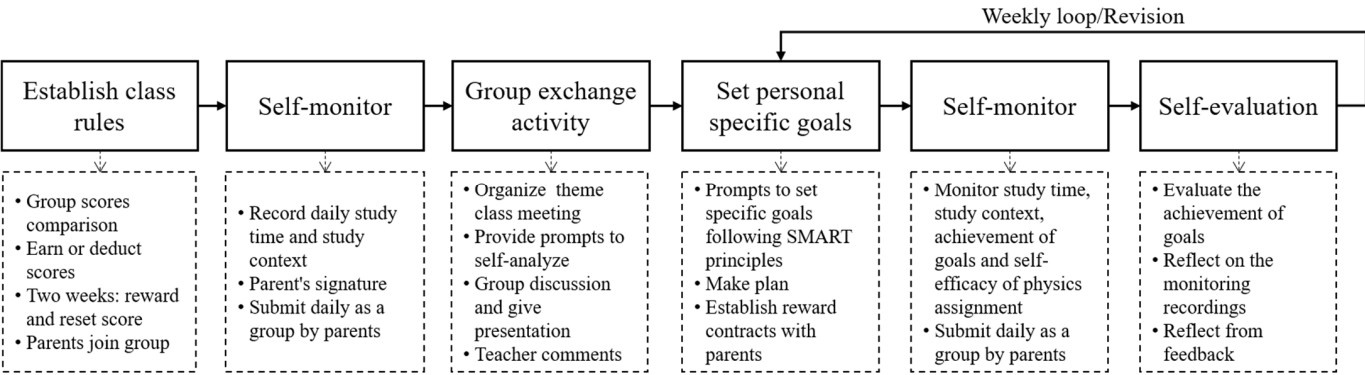

**Figure 4.** Implementation process of the first iteration.

In terms of goal setting and using monitoring forms, the instructor recommended using the monitoring form first. Thus, almost all students used monitoring forms (Figure A1) to self-monitor their study times and study contexts. Then, the students set weekly goals and made corresponding strategic plans based on the analysis of recordings and feedback (Figure A2). After that, the monitoring of self-efficacy and daily goal achievement was added to the subsequent daily self-monitoring form (Figure A1). Most parents were involved in assisting the children in goal setting and monitoring tasks at home via the process of obtaining a signature and establishing the contract. The monitoring and goal-setting forms were designed in Excel. The QQ group album was used to collect students' self-monitoring recordings in groups from Sunday to Thursday. Students also monitored self-efficacy three times on weekly physics tests.

Regarding social-interaction activities, a themed class meeting was conducted regarding how to learn more efficiently. Each group discussed online learning problems and regulatory strategies on Friday afternoon. All groups conducted internal exchanges and discussions to better prepare for sharing in class. However, the class meeting time was occupied by other school activities twice during this iteration.

5.1.2. Evaluation and Revision

First, based on our field observations and coding analysis of student interviews, the external incentive of group-score comparison motivated the students to participate (S1, S4). Each group had a group goal and aimed to obtain the reward. In the QQ group, the leader organized group members and encouraged them to complete learning tasks (S1, S2, S4–S9). One student said, "Our group leader is highly active, and her enthusiasm affects the activity of our entire group" (S4). However, several students did not submit the monitoring form in time (S6, S8). They treated the submission of the monitoring form as an assignment,

not a useful learning strategy. These students might have lacked an understanding of SRL skills and their importance.

Second, using monitoring forms to help students diagnose learning problems first and then set specific goals was a significant practice. Based on the observation of group presentations during class meetings, students easily identified their learning problems through the analysis of monitoring recordings and discussed possible solutions with group members. This was also confirmed in student interviews and the document analysis of monitoring forms. For example, one group recognized the role of the monitoring form in helping her identify sources of distraction: "I waste so much time because of distraction; I'm often distracted by my phone and often daze when I do my homework" (S1, S3). They further proposed solutions to overcome distraction, such as "having parents keep the phone until the homework is done" or "finding a quiet and undisturbed place to do homework".

We also found that, according to the document analysis, the prompts in the monitoring form needed to be dynamically adjusted as students' situations changed. For example, after analyzing the first week's monitoring forms, we found that almost all students had a fixed place to study. Therefore, we deleted the prompt "where you study" from the self-monitoring form, beginning in the second week. We also determined that the main distractions for students while studying were mobile phones (J5, J7, J19, J21, J23, J28, J29, J40, J41, J51, and J53), TV (J28, J40, and J53), and family members (J7 and J39). However, the frequency of distraction gradually decreased over time. Students started to realize the importance of time efficiency and were willing to adjust the learning environment and attempt to overcome distractions by setting goals. At the end of this iteration, the number of distractions per student was less than one per week. Furthermore, an obvious drawback of the monitoring form was that the daily individual forms were easy to lose and not conducive to longitudinal self-reflection. Students said they were "relatively unaccustomed to the Excel spreadsheets" (S7, S8) and that the forms were "a bit fragmented" (S7) and "easily lost" (S1, S3, S4).

Third, regarding social interaction, the groups discussed a certain topic every week, which helped them understand their problems and find solutions through sharing with others. However, the class meeting time was occasionally occupied by other activities, which led to less group communication and discussion.

Finally, regarding family–school co-education, all parents attended the goal-setting and monitoring activities. They signed the monitoring records and established goal contracts with their children, which enhanced communication and collaboration between parents, children, and the school.

Based on issues identified through feedback and analysis, we made the following revisions.

Rev. 1. Organize a systematic study session for SRL and related strategies.

Rev. 2. Adjust several prompts in the monitoring form based on students' experiences of using strategies.

Rev. 3. Adjust the use of Excel to record self-monitoring to make it more convenient for students.

Rev. 4. Adjust the method of task arrangement to obtain feedback from peers to reduce reliance on the class meeting time.

*5.2. Second Iteration*

5.2.1. Changes in Design

In this iteration, first, students were asked to attend a special session to stimulate their intrinsic motivation to acquire SRL skills (Rev. 1). Second, we changed the daily Excel monitoring form into a "*self-learning management notebook*" called a bullet journal [73], which was used to record journal entries covering planning, monitoring, and self-reflection (Rev. 3). In addition, the students determined the recording format and prompts based on their own needs (Rev. 2). We hoped this would help students internalize SRL skills by choice according to their needs as opposed to meeting a mandatory requirement. Finally,

peer-interaction activity was increased to once a week, online, for at least 5 min, and each group was asked to submit the main content of their discussion to obtain group scores (Rev. 4).

### 5.2.2. Implementation

This iteration lasted about 5 weeks. All students participated in the SRL class meeting in which the PI explained why SRL is important, how to use SRL strategies to learn more effectively, and the strategies implemented in their class. At the same time, the students started to use a self-learning management notebook to plan, self-monitor, and self-evaluate, combining the bullet journal method and the methods necessitated by their own actual situation (Figure A3). However, the students' journals were not submitted on time during the last 2 weeks, owing to a public holiday and a change of teachers.

Group discussion experiences and methods were shared in the second class-meeting. Six groups shared, and the teacher gave feedback. The next class meeting was a reading exchange and sharing session arranged by the school. The former head teacher re-engaged with the class and used the class meeting time to mobilize students, encouraging them to participate in the upcoming class sports and arts festival. Each student had to participate and had 2 weeks to prepare.

Furthermore, 5 min weekly online group discussion groups also started in the first week. The students' self-selected topics related to SRL or group scores, time, and platforms. However, owing to the holiday and the change of teachers, each group organized only two group discussions in this round. Most students were highly active in the discussions. Ten groups participated the first time and nine groups the second time.

### 5.2.3. Evaluation and Revision

First, the external incentive of group-score comparison continued to motivate the students to participate. When a group won a prize, they would celebrate in their QQ group. Moreover, after the PI held the special class meeting, students had a better understanding of why and how to regulate their learning, and they were more willing to invest time and effort in participating in SRL activities (S1, S2, S4).

Second, the change from the daily Excel form to a self-learning management notebook was more conducive to students' reflection. Several students said that learning management, monitoring, and reflection were more systematic and that entries could be accumulated (S2, S3, S8). They thought this change was particularly helpful (S1, S4, S6, S7, S9) because "the notebook can be accumulated and reflect my progress and regression for a long time" (S3). They also enjoyed such self-directed design (S4, S9) because "I can add my own arrangements such as playing basketball or running today, or other daily arrangements in my notebook" (S9).

The document analysis of students' journals showed that recording methods and monitoring strategies varied by student. Some students set goals, checked in every day (J1, J22, J40), and performed daily reflections (J5, J14, J25, J27, J28, J36, J37, J42, J47, J48), while some had no goals, leading to fewer reflections. In addition, some of the better student journals could not be seen by other students. Given factors such as holidays, the instructor gave limited guidance and feedback on the monitoring journals (S2). Thus, some students still had difficulty understanding how to use them to manage and monitor their learning.

Third, regarding the social interaction activity, the weekly group discussions helped remind the students to supervise each other to hand in homework and journals (S1, S2, S4–S9) and share some learning methods and SRL skills (S1, S9). "Our group leader is very responsible, and she always talks about the monitoring journals and reminds us to hand them in during group discussion" (S6). However, the teachers gave limited feedback on the group discussion (S2), so the students did not continue. Furthermore, the previous method of sharing and communication between groups took too much time, because the sharing of subsequent groups became repetitive.

Finally, regarding family–school co-education, over 93% of the parents continued to attend monitoring activities, according to the document analysis. This had slightly declined compared with the first iteration period, which might be attributable to the easing of the pandemic, when several parents resumed work and had little time to monitor their children. Conversely, this perhaps created an opportunity to transition from co-regulation to self-regulation. In the early stage, to increase communication between parents and children, the daily monitoring recordings were mainly uploaded by parents.

Therefore, we made the following revisions for the next iteration.

Rev. 5. Enhance the setting of goals to stimulate the application of students' SRL strategies.

Rev. 6. Strengthen teacher guidance and feedback on students' journals and online group discussions.

Rev. 7. Optimize the method of peer interaction and select a few groups for sharing.

Rev. 8. Adjust the submission method and platform for students' journals to meet the needs of parents returning to work.

Rev. 9. Enhance students' sense of self-efficacy and regulate the emotions associated with long-term home-based learning.

### 5.3. Third Iteration

#### 5.3.1. Changes in Design

In this iteration, students were asked to set weekly learning goals, self-monitor, and again provide feedback on their own goals (Rev. 5). Next, we optimized the class meeting process by incorporating teacher feedback on journals and group discussions (Rev. 6), selected group shares (Rev. 7), and a student live-broadcast activity (Rev. 9). Furthermore, we believed it was still necessary to submit monitoring recordings, because developing a habit takes time. Hence, the submission platform was changed to a more personalized teaching system. Students submitted their daily monitoring recordings by themselves, but parents could check and sign their journals (Rev. 8). Such measures not only reduced students' use of social software such as QQ to complete learning tasks but also further encouraged them to regulate their learning.

#### 5.3.2. Implementation

This iteration lasted about 6 weeks. The new instructor recognized these strategies and provided great support and optimization during the activities, as well as asking the students to submit their journals by themselves every day starting from the first week. Then, she tracked each student's submission in real time. Additionally, she resumed the requirement of setting goals every other week and daily monitoring, including tracing the goals in the self-learning management notebook (in fact, many students kept insisting on this), and provided feedback and demonstrations in the following week's class meeting.

Regarding the adjustment of class meeting time-allocation, the instructor allotted about 20 min for journal feedback and group discussion activities every week in the class meetings. The instructor then selected and displayed some of the better journals, giving specific praise such as an efficiency star, a standard star, and/or a target star. She also shared her own experience and provided other specific examples for students. Then, she gave more detailed comments to each group and presented the discussion processes and content of some excellent groups. At the same time, two groups were selected to share their groups' experiences with the class. As a result, the groups' 5 min weekly discussion became more substantial, and more groups were willing to participate to obtain basic group scores and bonus scores. In the remaining class meeting time, live broadcast activities arranged by the students were conducted every week, such as the teaching of painting, crafts, and IT skills. Students enjoyed their work achievements.

### 5.3.3. Evaluation and Revision

The evaluation in this iteration was a summative assessment that focused on evaluating the feasibility and effectiveness of the implementation mechanism, including all instructional design principles and students' improvement of SRL skills.

After three rounds of iteration, we found that the integrated strategies of stimulating motivation, implementing individual tangible records, and engaging in group discussion activity were continuously effective. According to data from the monitoring journals submitted by students and the group discussion records, 81 and 91% of students constantly engaged in writing in their monitoring journals and in group discussions, respectively. Nearly 80% indicated that using monitoring journals was a helpful way to manage learning and improve efficacy and reflection, which was also recognized by the teacher and more than 80% of the parents (Appendix Tables A3 and A4). Through further thematic analysis (Appendix Table A2), students indicated that they "experienced the fulfillment of self-directed learning" (S2), "became more self-disciplined" (S6), and would "continue to persevere during summer vacation or later" (S1, S2, S4–S9). Furthermore, more than 70% of the students thought that weekly group-discussion activities could facilitate their studies, and more than 80% were willing to share problems during online discussions, which could also "make up for the loneliness of online learning" (S2).

Regarding the instructor's feedback and guidance regarding SRL skills, students felt the instructor's examples and reviews of monitoring journals and goal-setting records were highly useful (S1, S2, S4–S8). "At first, I did not know what a monitoring form was, but after her analysis, it became clear what I should do" (S4). Students better understood how to design the journals (S1, S2, S6, S7). Further, the instructor's feedback of awarding star ratings during class meetings motivated students to more actively engage in SRL activities (S2, S3), which enabled them to achieve more transformation and growth (S1). The teacher noted that "a positive feedback loop was being formed (T)".

Regarding family–school co-education, the online survey results (see Appendix Table A4) revealed that over 90% of the parents proactively communicated with children or helped them use journals to analyze learning problems when signing them. This suggests that having parents sign the monitoring journals facilitated communication and collaboration between parents and children. Nearly 70% of the parents participated in guiding or supervising their children in setting goals. They mentioned that the monitoring journal was a good way to "know their children's learning status" (P9, P15, P22, P28, Pi2), "supervise children to finish homework" (P5, P7, P8, P12, P16, P18, P23, P31, Pi1, Pi3), and help children engage in more effective reflection (P11, P17, Pi3). Furthermore, more than 70% of the parents were satisfied with their children's self-disciplinary performance during online learning.

In summary, the three iterations of the program supporting home-based SRL among secondary school students gradually improved the implementation mechanism, and the changes in self-regulation awareness and behaviors suggested the improvement of students' SRL skills (e.g., their self-reflection ability). The proposed implementation mechanism, including instructional sequence, corresponding resources, SRL activities, and class-incentive rules of practice, as shown in Figure 3, proved to be feasible and effective. Additionally, we made several changes to the design of the implementation mechanism based on our evidence, as shown in Figure 5.

Appendix A Table A5 provides a summary of the design, implementation, and evaluation activities in the prototyping process.

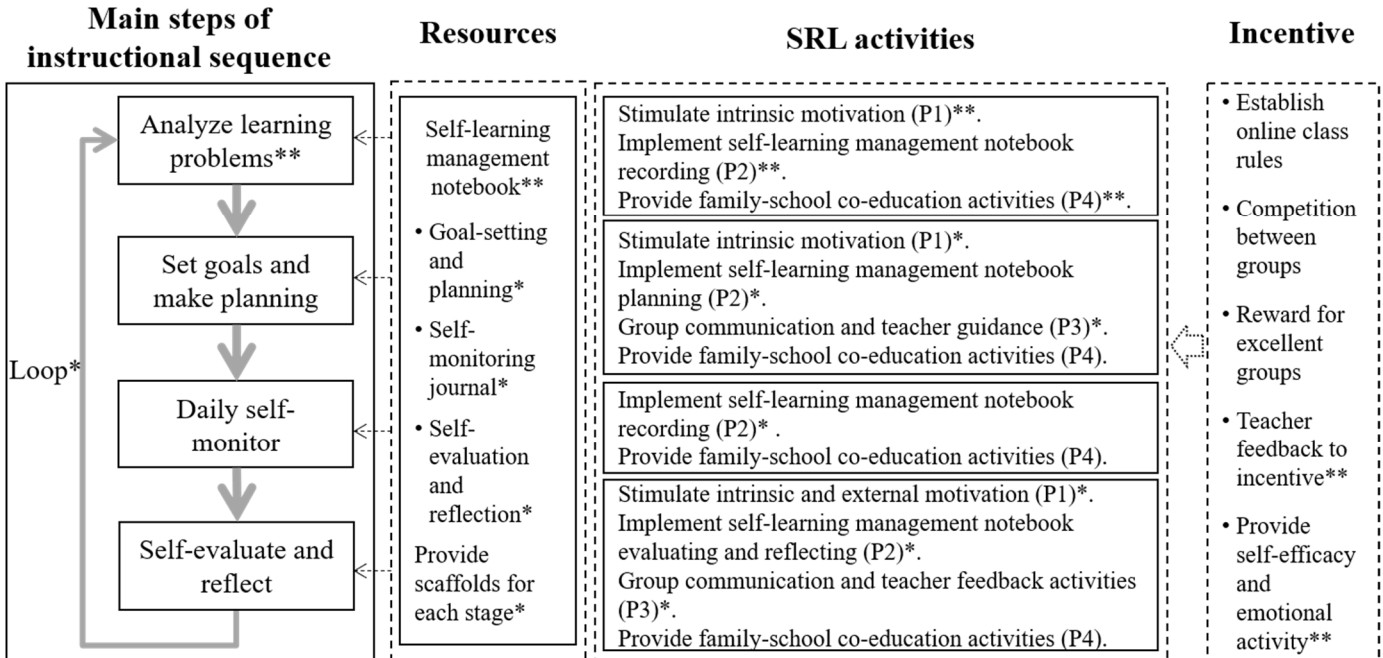

**Figure 5.** Revised instructional design of the implementation mechanism. Note: the modified part is marked with *, and the added part is marked with **.

## 6. Discussion

Using the educational-design research approach, we aimed to design a feasible and effective implementation mechanism for practices that support secondary school students' SRL in home-based learning environments. Here, we discuss the key findings according to the following four aspects.

### 6.1. The Proposed Mechanism Worked Well in Home-Based Learning Environments and Could Be Applicable to Other Learning Contexts

Our findings demonstrate that integrating SRL strategies such as goal setting, planning, self-monitoring, and self-evaluation through our implementation mechanism can effectively develop secondary students' SRL skills and enhance learning performance in home-based learning environments. This aligns with similar findings in higher education [7,74] and secondary education [10]. Home-based learning, conducted primarily online with limited teacher supervision [74], offers an opportunity for students to improve their self-regulation skills and foster family–school collaboration [62,75]. Feedback from students and parents indicates that the instructional sequence, supported resources, and incentivized SRL activities have a positive impact on students' planning, monitoring, and effective use of study time. These insights, although derived from a specific study on home-based SRL during a pandemic in China, have broader implications for future online or blended learning scenarios that require self-regulatory skills [2,10]. Considering the increasing prevalence of smart devices and the potential for global emergencies like COVID-19, hybrid learning environments may become more common in secondary schools. Therefore, our refined implementation mechanism, which involves teacher involvement in students' SRL and promotes family–school collaboration, serves as a valuable reference for future flexible or e-learning scenarios in secondary school education in the post-pandemic era.

### 6.2. Family–School Co-Education Plays a Significant Role in the Development of Self-Regulated Learning Skills in Secondary School Students

Secondary school is a critical period for developing SRL skills [76]. In home-based learning environments, family plays an essential role in facilitating students' behavioral engagement and fostering the development of SRL skills [25–27]. Previous studies suggest

that equipping students with SRL skills should be a major goal of school education [77], aiming to fill the gap in initial SRL levels caused by different family backgrounds [78]. However, most secondary schools do not have systematic SRL programs for students [36]. Therefore, the dilemma is that most teachers and parents lack the experience and knowledge of SRL needed to guide students, even though they recognize the importance of SRL for lifelong learning [17,76]. The acquisition of self-regulatory skills is not only inseparable from adult guidance but also requires parents and teachers to give more autonomy to children [63], thus maximizing the acquisition of SRL skills via family–school co-education [62]. This study revealed that providing teachers and parents with appropriate scaffolding and participation opportunities can build a communication bridge between parents, teachers, and children, thereby maximizing the promotion of students' SRL. Furthermore, our implementation mechanism provides ready-to-use guidance and resource packs for teachers and parents.

*6.3. Social Interaction with Teachers and Peers Motivates Engagement in Self-Regulated Learning Activities*

Students' lack of motivation to participate in SRL activities might be the greatest challenge in implementing this intervention. It is important to keep students continuously motivated to further engage in SRL activities, especially in online settings [7,74,79]. In this study, we attempted to stimulate students' internal and external motivations from multiple perspectives. Among these perspectives, we found that the interaction between teachers, students, and peers had a great effect on stimulating students' continuous motivation to participate. This is consistent with previous studies showing that social feedback from a teacher or peers is essential for developing students' emulative conduct and facilitating further engagement [37,59,60]. Especially in the case of beginners, students might easily lose interest if they do not receive social encouragement and guidance [80]. Two students mentioned that they were particularly unwilling to self-record at first, but with guidance from teachers and encouragement from peers, they gradually became willing to participate and found it useful. At the same time, after the third iteration, students talked about the effects of peer supervision and teacher feedback and guidance, further showing that social interaction can facilitate students' further engagement and the transition from external motivation to internal self-guidance and self-regulation [12].

*6.4. The Paper–Pencil Approach Is Preferable for Creating Tangible Records*

In this study, the tangible records changed from electronic spreadsheets (i.e., Excel v2019) to paper-based notebooks (i.e., self-learning management notebooks) [36]. Some might view handwritten records as a step backward. However, many studies have shown that handwriting is increasingly important in today's information age because it can stimulate the brain more effectively through touch and promote deeper thinking and understanding [73]. One student mentioned that "paper is a material that you can feel and see, and writing is impressive". Surprisingly, several students said they would continue to use a self-learning management notebook. In follow-up interviews the next semester, some even reported using sticky notes to make lists every day without anyone requesting them. This confirms that the paper–pencil approach can help secondary students deepen their self-cognition and acquire self-regulatory skills when learning online. In addition, the paper–pencil approach does not impose any requirements on students' families and is more convenient for students than is digital recording. We cannot ignore the advantages of technology, such as timely data collection and analysis [81], but using digital technologies could also potentially increase students' cognitive loads. Furthermore, owing to concerns about exposure to inappropriate content and excessive gaming, many Chinese parents disapprove of their children's use of mobile phones, thereby rendering complete reliance on digital devices unfeasible at present.

### 7. Conclusions

Based on our results, we propose seven refined instructional design principles for developing secondary school students' SRL skills. Some of them have enriched connotations and have been made more specific (P1–P5), and some have been added based on our findings (P6, P7). In addition, these principles are an integrated framework, but can also be further adjusted according to the changes in the educational context.

P1. Stimulate the intrinsic motivations (e.g., analyzing learning problems, giving the rationale behind strategies, and setting goals continuously) and external motivations (e.g., group competition) of students to engage in SRL activities.

P2. Constantly implement individual tangible records as tools for the practice of students' planning, monitoring, and evaluating skills, gradually giving them autonomy in the management their acquisition of SRL skills and adjusting the scaffolds according to dynamic changes in their technical level of self-regulation.

P3. Organize regular peer-communication activities so they can share and obtain peers' feedback on the SRL learning process.

P4. Provide individualized feedback, social encouragement, and guidance on common issues through teachers to facilitate students' work within the learning process.

P5. Create opportunities for parents to participate in the SRL skill learning process and provide them with certain guidance and scaffolding.

P6. Follow the general sequences of "learning problem analyzing–goal setting and planning–daily monitoring–evaluating" to develop SRL skills.

P7. Be flexible about the four aspects of the mechanism and follow the principle of gradual progress, starting with less progress and gradually increasing.

However, several limitations of the current study should be noted when interpreting the aforementioned SRL design principles. First, the study was conducted in a top secondary school in Wuhan during a pandemic, which could lead to the findings being less generalizable to other contexts, such as disadvantaged schools. Second, this study involved one eighth-grade class with 58 participants. Therefore, the feasibility and effectiveness of the implementation mechanism remain unknown for other grades. Third, students' SRL processes were explored primarily via self-perspectives and self-recording, while their social forms of regulation and real-time regulation behaviors were not observed. Thus, we did not capture whether the perspectives and journals completely reflected students' actual thoughts. Consequently, future research should explore how to better observe real-time regulatory behavior and provide personalized feedback based on timely data analysis through an information technology approach.

**Author Contributions:** Conceptualization, funding acquisition, supervision, methodology, writing—original draft: M.Z.; conceptualization, investigation, supervision, resources, formal analysis, writing—original draft: Q.Z.; conceptualization, methodology, supervision, writing—reviewing and editing: Q.W.; material preparation, investigation, formal analysis, writing—original draft: Y.Y.; material preparation, investigation, formal analysis: L.L.; material preparation, investigation, formal analysis: W.G.; conceptualization, methodology, supervision, writing—reviewing and editing: H.L. All authors have read and agreed to the published version of the manuscript.

**Funding:** This research was funded by Collaborative Innovation Center for Informatization and Balanced Development of K-12 Education by MOE and Hubei Province, grant number xtzdwt2022-001.

**Institutional Review Board Statement:** This study was conducted in accordance with the Declaration of Helsinki and approved by the Ethics Committee of Central China Normal University.

**Informed Consent Statement:** Informed consent was obtained from all subjects involved in the study: they could withdraw from the study at any time without penalty, and their personal identifiable information would be kept anonymous in all publications and presentations.

**Data Availability Statement:** The data that support the findings of this study are openly available in Mendeley Data at https://doi.org/10.17632/kn2c9x2k3x.1 (accessed on 28 January 2024).

**Acknowledgments:** We greatly appreciate all the participants' collaboration.

**Conflicts of Interest:** The authors declare no conflicts of interest.

## Appendix A. List of Appendix Tables

**Table A1.** First cycle coding methods used for data reduction.

| Method | Definition | Operation | Example Codes |
|---|---|---|---|
| Structural coding | Conceptual phrases representing topics of inquiry to address specific research questions. | Structural codes consisted of a list of preconceived evaluation questions, such as those inquiring into the experience of using monitoring forms, and as to their perceived effects. | MONITORING FORM, LEARNING GROUP, TEACHER FEEDBACK, PARENT COMMENT |
| In vivo coding | Actual words or phrases used by participants, also known as "verbatim coding". | In vivo codes used participants' own words from the interviews to describe their learning experiences to ensure any interpretations and conclusions were grounded in data. | "SCATTERED", "CIRCUMSCRIBED", "MORE CLARITY", "WEEKLY SUMMARY", "SUPERVISE" |
| Magnitude coding | Alphanumeric or symbolic codes to indicate intensity, frequency, direction, presence, or evaluation. | Magnitude codes were combined with other codes to evaluate instructional design features: the frequency, intensity, and overall opinions. | POS = POSITIVE, NEG = NEGATIVE, NEU = NEUTRAL; EXTREMELY (E), DETERMINE (D), POSSIBLE (P) |
| Value coding | Codes reflecting participants' values, attitudes, and beliefs, or representing their perspectives or worldviews. | Value codes directly address research questions regarding valued and not-valued design features. Some value codes were determined a priori, and some were constructed during data coding. | Values: HEALTH, HABIT, BE LIVELY<br>Attitudes: USEFUL, HUMOROUS, PRETTY GOOD, NOT ALONE<br>Beliefs: ENRICHED, COMPLETE AS SCHEDULED |
| Evaluation coding | Codes that assign judgments about merit or worth, or the significance of programs or policies. | Evaluation codes were extracted based on researcher judgments and participant comments and used with other coding methods to address evaluative inquiries regarding the instructional design features and effects. | IMPROVE EFFICIENCY, + "PRESIST EVERYDAY", − OCCASIONALLY FORGET, REC (recommendation): WRITE A DAILY THOUGHT |
| Versus coding | Dichotomous or binary codes applied to a segment of data, in directions conflicting with each other. | Versus codes highlight participants' different or contradicting views on their learning experiences; also indicate conflicting findings from pilot and field tests. | NOTEBOOK VS A FORM, ESTIMATED TIME VS ACTUAL TIME, PAST VS PRESENT, USEFUL VS GENERAL |

**Table A2.** A summary of emergent themes and frequencies regarding the perceived usefulness of each principle for students and parents.

| Themes | Example Codes | Students | |
|---|---|---|---|
| | | Case | Code |
| Stimulate motivation (2 themes, 54 codes) | | | |
| Motivated to engage in SRL activities by peers and teacher | Active group leader, like to add points, everyone is very positive, reminder from group leader not to deduct points, inspired by star review, etc. | 7 | 24 |
| Perceived usefulness and continuing to use | Monitoring journal is very useful, continue to use consistently, teacher feedback is very useful, group discussion helps me solve problems, etc. | 9 | 30 |

**Table A2.** *Cont.*

| Themes | Example Codes | Frequency | | | |
|---|---|---|---|---|---|
| | | Students | | Parents | |
| Implement individual tangible records (6 themes, 84 codes) | | Case | Code | Case | Code |
| Stay organized and on top of daily homework with ease | Know daily homework, seeing whether completed all assignments, not easy to forget special homework, knowing what to do, more organized, etc. | 4 | 11 | 10 | 10 |
| Enhance learning productivity and gain more free time with improved homework efficiency | Improve homework efficiency, a lot of extra time to do things you can't do before, improve learning efficiency, etc. | 3 | 6 | 3 | 3 |
| Facilitate the identification and addressing of learning issues through reflection | Helped me detect procrastination, discovering issues with insufficient time, find areas for adjustment and improvement, see if the goal has been achieved, reflect one's progress and decline, etc. | 4 | 16 | 2 | 2 |
| Improve time management and achieve better study habits | Better evaluation of time, realizing that there is not enough time, plan and arrange, time management, help arrange study time reasonably, etc. | 3 | 6 | 7 | 8 |
| Promote self-discipline and achieve academic goals with monitoring | Monitoring role, supervise the completion of homework on time, enhancing self-discipline, supervise children to achieve goals, etc. | 1 | 1 | 11 | 16 |
| Experience the satisfaction of self-directed learning and personal growth | Seeing one's own progress, daily learning becomes more fulfilling, it's my own, more self-directed learning, independently complete on time, etc. | 3 | 3 | 2 | 2 |

| Themes | Example Codes | Students | |
|---|---|---|---|
| Group discussion activity (5 themes, 38 codes) | | Case | Code |
| Increase accountability and engagement through mutual supervision and reminders | Remind to submit monitoring journal, supervise the submission of homework, remind to attend class seriously, etc. | 8 | 13 |
| Collaborate and make progress together by addressing challenging questions and solving problems. | Help to solve problems, discuss the question that cannot be done, make progress together, etc. | 5 | 13 |
| Learn from classmates' designs to improve one's own monitoring journal design | Knowing own shortcomings, drawing on the design of monitoring journal, etc. | 4 | 8 |
| Increase efficiency through synchronous task completion and adherence to agreements | Increase efficiency, do the agreed task synchronously, etc. | 1 | 2 |
| Make up for the loneliness of online learning | Feel like being with everyone, feel not alone, etc. | 2 | 2 |

| Themes | Example Codes | Students | |
|---|---|---|---|
| Instructor's feedback and guidance (4 themes, 14 codes) | | Case | Code |
| Assist in designing and improving monitoring journal | Develop the content of monitoring journal, better at designing journals, give me a lot of inspiration, seeing designs from other classmates, etc. | 4 | 6 |
| Help students identify areas for improvement and define action steps for progress | Know where to improve, clarify what to do, etc. | 2 | 2 |
| Increase individual fulfillment, motivation, and participation in SRL activities | Make me feel more fulfilled, more motivated, more active to attend group discussions, etc. | 2 | 5 |
| Enable students to achieve greater transformation | The transformation is greater than before, etc. | 1 | 1 |

| Themes | Example Codes | Parents | |
|---|---|---|---|
| Family-school co-education (2 themes, 11 codes) | | Case | Code |
| Facilitate parental supervision | Effectively supervise when signing, urge children to achieve goals, urge completion of homework, etc. | 8 | 8 |
| Better knowing of children's learning status | Knowing children's learning status, knowing the degree of completion of homework, etc. | 3 | 3 |

**Table A3.** The results of students' attitudes towards the designs.

| Students (N = 51): | | | M | SD |
|---|---|---|---|---|
| 1. | | The use of self-learning management notebook helps to manage my learning. | 3.90 | 0.900 |
| 2. | | Daily self-monitoring journal is helpful for me to improve my learning efficiency. | 3.90 | 0.781 |
| 3. | | I often reflect on my learning based on my monitoring journal. | 3.94 | 0.835 |
| 4. | | Weekly group discussion activities for 5 min are helpful to my study. | 3.71 | 0.986 |
| 5. | | I share my problems with peers when discussion online so we know what we are struggling with and how to solve our problems. | 4.02 | 0.969 |

**Table A4.** The results of parents' attitudes towards the designs.

| | Parents (N = 56): | M | SD |
|---|---|---|---|
| 1. | I try to communicate with my child initiatively or help him use journals to analyze learning problems when signing the monitoring journal every day. | 4.29 | 0.731 |
| 2. | I participated in guiding or supervising my child to set goals. | 4.21 | 0.909 |
| 3. | The using of self-monitoring journal helped my child's study. | 4.04 | 0.990 |
| 4. | I am satisfied with my child's self-discipline performance during online learning. | 3.57 | 1.006 |
| 5. | After returning to school, I hope to continue to use self-monitoring journal in the class. | 3.93 | 1.042 |

**Table A5.** An overview of the prototyping process.

| | First Iteration | Second Iteration | Third Iteration |
|---|---|---|---|
| Design | <ul><li>The instructor established online class rules.</li><li>Students used Excel self-monitoring form.</li><li>Students set weekly goals, monitored and self-evaluated continuously.</li><li>The instructor created some peer communication and discussion opportunities to facilitate self-reflection and SRL learning.</li></ul> | <ul><li>The instructor organized a special session to explain what SRL is.</li><li>The instructor established an online group discussion system.</li><li>Students used self-learning management notebooks to plan, monitor, and reflect.</li><li>The recordings of some prompts were adjusted, making them optional.</li></ul> | <ul><li>The instructor strengthened feedback on monitoring journals and group discussion in class.</li><li>Two groups communicated and shared in each class meeting.</li><li>Students' goal setting, monitoring, and feedback on the goals were submitted.</li></ul> |
| Implementation | <ul><li>It lasted about four weeks.</li><li>Online class rules were established, and group competition activities were hold.</li><li>Parents joined the group.</li><li>Students started to self-monitor and submitted daily recordings to the QQ group album with parents.</li><li>The instructor guided students to monitor self-efficacy of physics quizzes, self-evaluate and set goals.</li><li>Class meetings were used for peer communication and feedback, while sometimes otherwise occupied.</li></ul> | <ul><li>It lasted about five weeks.</li><li>An online themed class meeting about SRL was conducted and five-minute group online discussions once per week were organized.</li><li>Students started to use the self-learning management notebooks to regulate their learning in their own styles, and parents attended.</li><li>The May Day holiday and the replacement of the class head teacher led to delayed submission and feedback.</li></ul> | <ul><li>It lasted almost six weeks.</li><li>The instructor showed samples and gave detailed feedback and guidance on students' daily recordings and group commutations.</li><li>Students began to use Zhixuewang to submit various assignments, including their own journals.</li><li>The instructor resumed the requirement of setting weekly goals; monitoring and feedback and requirement of parent's signature continued.</li><li>Five-minute group online discussion once a week was continued.</li></ul> |
| Evaluation | <ul><li>Excel forms for recordings were not convenient for continuous self-reflection.</li><li>Students could choose the learning environment and overcome distractions more consciously.</li><li>Peer communication could not be promoted by class meetings alone.</li><li>How to promote students' systematic cognition of SRL?</li></ul> | <ul><li>Instructor's guidance and feedback on using journals effectively and group online discussions were insufficient.</li><li>Some groups, instead of all, should share in class meetings.</li><li>The submission method of journals needed to be adjusted.</li><li>How to strengthen students' goal setting to stimulate the utility of SRL strategies?</li></ul> | <ul><li>Students' awareness of time and self-regulation were enhanced.</li><li>Students thought instructor's feedback was highly useful.</li><li>Peers' interaction promoted mutual supervision and reflection.</li><li>Parents' participation facilitated communication and collaboration between parents and children.</li></ul> |

## Appendix B. List of Appendix Figures

Added after setting weekly goals

| Date | Assignment | Time started | Time spent | Study context | | | Self-efficacy | Today's goal achievement | Parent's signature |
|---|---|---|---|---|---|---|---|---|---|
| | | | | Where? | With whom? | Distractions? | | | |
| | | | | | | | | | |
| | | | | | | | | ☐ Yes | |
| | | | | | | | | ☐ No | |

**Figure A1.** Design of the self-monitoring form.

### My goal setting and planning form

| Self-analyze and evaluation | | | Week | Set goals and make strategic plan | |
|---|---|---|---|---|---|
| NO. | Process monitoring | Feedback | | Goals | Strategic planning |
| 1 | Do math (physics) mid-term training manual at 13:30~13:55 every day | Physics Weekly Test =83 Self-efficacy =84 | 1 | Do not play with mobile phones during math homework at night | Put electronic products outside when studying |
| 2 | Start to make up yesterday's homework at 6:30 in the morning, endorse the words | Self-efficacy=5 | | Complete the task within each specified time | Set an alarm to increase the sense of urgency when studying |
| If the goals are not reached one day next week, the consequences I need to bear are: | | | Don't touch the phone for 3 days | | |

**Student signature:** ▓▓▓▓

**Parent's signature:** ▓▓▓▓

**Date:** Mar. 21, 2020

**Figure A2.** A student's goal-setting and planning form.

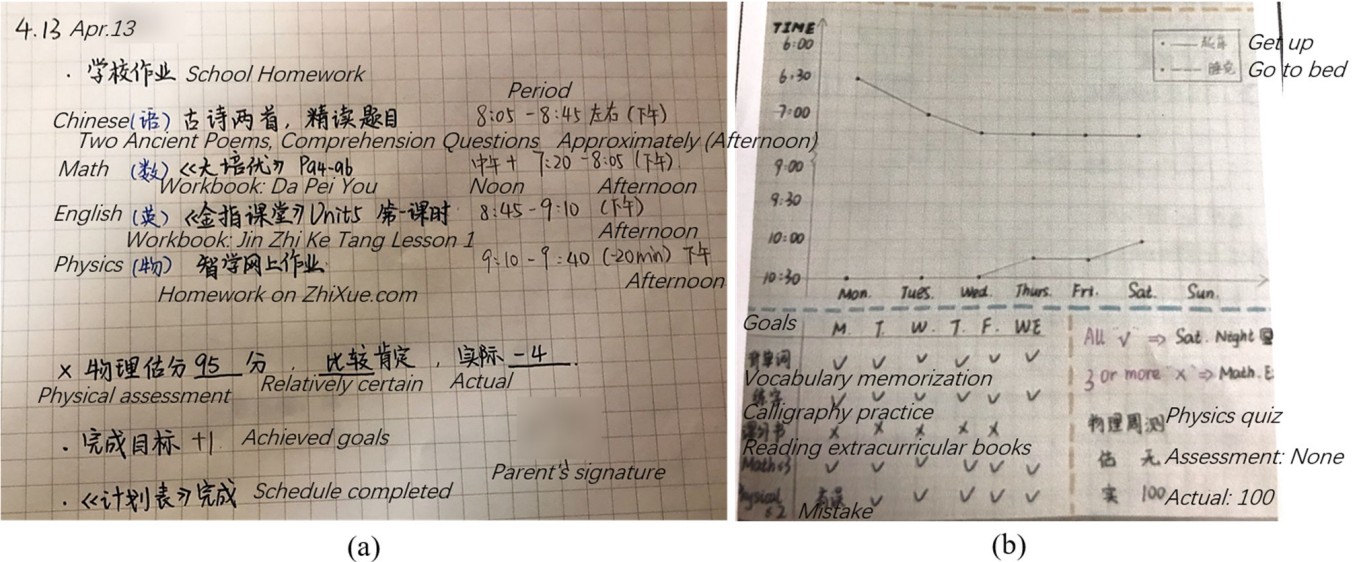

(a)                    (b)

**Figure A3.** Examples of students' journals: (**a**) a student's daily recording, and (**b**) a student's weekly goals and trace recording.

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
