# Peer review of "Supporting Home-Based Self-Regulated Learning for Secondary School Students: An Educational Design Study"

_sustainability, doi:10.3390/su16031199_

Round 1

Reviewer 1 Report

Comments and Suggestions for Authors

The article is very well designed with all sequences. All the descriptions are good enough for the reader´s idea about the research. The article is related with sense and contribution to the field of education. The references are appropriate.

Author Response

Dear Reviewer:

Thank you for your encouraging comments. We are glad that the value of our study was recognized by you.

Reviewer 2 Report

Comments and Suggestions for Authors

This is a well-written paper. I have minor points for the authors.

1.  Please add the theoretical support to the four principles (e.g. motivation)in SRL development.

2.  It’s not very common to give a special training on SRL in high school classroom teaching, thus, the participants may not be representative. I suggest the current study should present more details about the students who lack SDL skills

3.  Please do a careful copyediting.

Comments on the Quality of English Language

This is a well-written paper. I have minor points for the authors.

1.  Please add the theoretical support to the four principles (e.g. motivation)in SRL development.

2.  It’s not very common to give a special training on SRL in high school classroom teaching, thus, the participants may not be representative. I suggest the current study should present more details about the students who lack SDL skills

3.  Please do a careful copyediting.

Reviewer 3 Report

Comments and Suggestions for Authors

Paper deals with important problem of implementing effective ways of home-based learning for secondary school students in conditions that includes lack of previous learning experience and low self-regulated learning process. Topic is very important for effectiveness of modern education and this study is   well designed and backed with relevant sources. In other tor study to be more approachable and useable for educational theorist and practitioners some improvements should be made.

-          Title should be shortened. Part “An Educational Design Study” may be omitted from title.

-          In abstract key results should be explained in more detail and more specifically

-          Limitations of the study should be mentioned in abstract

-          Some important implications of results should be mentioned in abstract

-          Main text is longer than necessary on should be shortened where possible

-          Some figures and tables could be moved in appendixes section (for example fig. 5,6,7 …)

-          Discussion and conclusion section should be separated and shortened where possible

-          Some corrections regarding language and style can be made in other for  paper to be more comprehensible and approachable to experts but also to students and parents.

Comments on the Quality of English Language

/
